# The Long Non-Coding RNA MALAT1 Modulates NR4A1 Expression through a Downstream Regulatory Element in Specific Cancer Cell Types

**DOI:** 10.3390/ijms25105515

**Published:** 2024-05-18

**Authors:** Sara Wernig-Zorc, Uwe Schwartz, Paulina Martínez-Rodríguez, Josefa Inalef, Francisca Pavicic, Pamela Ehrenfeld, Gernot Längst, Rodrigo Maldonado

**Affiliations:** 1Regensburg Center for Biochemistry [RCB], Universität Regensburg, 93053 Regensburg, Germany; sara.wernig-zorc@ccri.at (S.W.-Z.); 2St. Anna Children’s Cancer Research Institute, 1090 Vienna, Austria; 3NGS Analysis Center, Biology and Pre-Clinical Medicine, Universität Regensburg, 93053 Regensburg, Germany; 4Programa de Doctorado en Ciencias, mención Biología Celular y Molecular Aplicada, Universidad de La Frontera, Temuco 4811230, Chile; paulinaconstanza.martinez@ufrontera.cl; 5Institute of Anatomy, Histology, and Pathology, Faculty of Medicine, Universidad Austral de Chile, 5090000 Valdivia, Chile; ingridehrenfeld@uach.cl (P.E.); 6Center for Interdisciplinary Studies of the Nervous System [CISNe], Universidad Austral de Chile, 5090000 Valdivia, Chile; 7Facultad de Medicina y Ciencias, Universidad San Sebastián, 5110246 Valdivia, Chile

**Keywords:** chromatin-associated lncRNAs, MALAT1, NR4A1, chromatin accessibility, breast cancer

## Abstract

Long non-coding RNAs (lncRNAs) have been shown to modulate gene expression and are involved in the initiation and progression of various cancer types. Despite the wealth of studies describing transcriptome changes upon lncRNA knockdown, there is limited information describing lncRNA-mediated effects on regulatory elements (REs) modulating gene expression. In this study, we investigated how the metastasis-associated lung adenocarcinoma transcript 1 (MALAT1) lncRNA regulates primary target genes using time-resolved MALAT1 knockdown followed by parallel RNA-seq and ATAC-seq assays. The results revealed that MALAT1 primarily regulates specific protein-coding genes and a substantial decrease in the accessibility downstream of the *NR4A1* gene that was associated with a decreased *NR4A1* expression. Moreover, the presence of an NR4A1-downstream RE was demonstrated by CRISPR-i assays to define a functional MALAT1/NR4A1 axis. By analyzing TCGA data, we identified a positive correlation between NR4A1 expression and NR4A1-downstream RE accessibility in breast cancer but not in pancreatic cancer. Accordingly, this regulatory mechanism was experimentally validated in breast cancer cells (MCF7) but not in pancreatic duct epithelial carcinoma (PANC1) cells. Therefore, our results demonstrated that MALAT1 is involved in a molecular mechanism that fine-tunes NR4A1 expression by modulating the accessibility of a downstream RE in a cell type-specific manner.

## 1. Introduction

Histones, chromatin-associated proteins, and RNA play essential roles in fine-tuning gene expression. Among them, non-coding RNAs have been identified as key factors modulating nuclear architecture, genome-wide chromatin interactions, gene expression, and genomic integrity [1]. Long non-coding RNAs (lncRNAs) are transcripts of at least 200 nucleotides (nt) in length transcribed by RNA polymerase II; in most cases, they are spliced, 3′ polyadenylated, and 5′ capped by 7-methyl guanosine [2]. LncRNAs function as cis- or trans-acting transcriptional regulators depending on their functional interactions with chromatin, chromatin-associated factors, and nuclear ribonucleoprotein particles [3]. According to information from different databases, almost 57,000 lncRNA genes have been found to generate more than 127,000 different transcripts from the human genome [4]. The expression of lncRNAs is highly tissue- and cell-type specific, and their regulation of stem cell differentiation and renewal, in combination with specific protein partners, demonstrates their remarkable transcriptional regulatory potential [5]. Lineage engagement for adipose, skeletal, cartilage, muscle, neuronal, and skin cell differentiation, in addition to embryonic stem cell pluripotency, has been described to be guided by different lncRNAs, such as PU.1 AS, Msx1as, HOTTIP, H19, HOTAIRM1, TINCR, RoR, Xist, Airn, and Firre, among others [6,7,8]. Thus, lncRNAs are associated with the onset and progression of cancer by regulating gene expression at the transcriptional and posttranscriptional levels, affecting intermediary metabolism and cell signaling [2,9,10,11].

The metastasis-associated lung adenocarcinoma transcript 1 (MALAT1) is one of the most abundant lncRNAs in human cells and is localized mainly on nuclear speckles [12]. In humans, MALAT1 is an 8 kb single-exon transcript expressed at chromosomal region 11q13, and studies on its mouse homolog have shown that RNaseP-RNaseZ posttranscriptional processing results in the retention of a long transcript on nuclear speckles and a 61 nt small RNA with a tRNA-like structure [13,14]. Intriguingly, after RNase processing, the 3′ end of the long MALAT1 transcript folds to generate a blunt-ended triple helical structure that stabilizes the transcript through protection from 3′-to-5′ exonucleases [15,16].

The role of the lncRNA MALAT1 has been studied by RNA–RNA interactions using RNA–antisense purification (RAP) and sequencing experiments in mouse embryonic stem cells, which revealed that MALAT1 interacts mainly with pre-mRNAs, such as arginine/serine-rich phosphoproteins, through proteins involved in alternative splicing [17]. Additionally, the function of MALAT1 has been studied by capture hybridization analysis of RNA targets (CHART) in MCF7 breast cancer cells, which was found to interact with DNA at numerous genomic sites, preferentially associated with the coding region and termination or polyadenylation sites of transcriptionally active and spliced genes [18]. Then, by combining mass spectrometry with the CHART assay (CHART-MS), the preferential interaction of MALAT1 with RNA processing factors, such as nuclear speckle and paraspeckle proteins and transcriptional regulators, was revealed [18].

Alterations in MALAT1 expression and/or function have been associated with different types of cancer. MALAT1 has been shown to act as a tumor suppressor or a tumor promoter, depending on the cancer type examined, or even to have opposite roles in the same cancer type [9,19,20]. These discrepancies could be explained by the diverse effects of MALAT1 downregulation on different cancer cells, where genome-wide transcriptomic changes have been reported. However, studies on the time-dependent modulation of MALAT1 expression and how this alteration in gene expression and chromatin accessibility affect gene-specific mechanisms of transcriptional regulation in cancer are still scarce.

In this study, we performed a time-resolved modulation of MALAT1 expression with locked nucleic acid (LNA) Gapmers followed by RNA-seq to define primary transcriptional targets in HeLa cells as a model system. In parallel, chromatin accessibility changes were analyzed by ATAC-seq, which revealed general changes and specific and considerable loss of accessibility for an element downstream of the nuclear receptor subfamily 4 Group A Member 1 (*NR4A1*) gene associated with its downregulation. Moreover, using CRISPR-i technology, we determined a functional axis in which MALAT1 modulates the accessibility of the NR4A1-downstream RE regulating *NR4A1* expression (the NR4A1/MALAT1 axis). NR4A1 is a transcription factor that is dysregulated in different cancer types, involved in T-cell dysfunction, and associated with the development of fibrotic diseases [21,22,23,24,25,26]. Therefore, to understand the role of this axis in cancer, we correlated the chromatin accessibility of an NR4A1-downstream RE and the expression of the *NR4A1* gene in TCGA data. Interestingly, we found cancer type-specific correlations, such as in breast cancer, but no correlation was observed in the pancreatic cancer data. Then, we experimentally evaluated the effect of the NR4A1/MALAT1 axis on specific cancer types through MALAT1 downregulation and CRISPR-i assays. The results showed that the *NR4A1* gene was specifically regulated by MALAT1 in MCF7 breast cancer cells but not in pancreatic ductal adenocarcinoma PANC1 cells. Our findings delineate the molecular mechanism of MALAT1 that fine-tunes the expression of specific cancer modulators, such as NR4A1, by regulating the chromatin accessibility of an NR4A1-downstream RE.

## 2. Results

### 2.1. MALAT1 Primarily Regulates Protein-Coding Genes

To determine the primary transcriptional targets involved, we performed a time-dependent modulation of MALAT1 expression by transfecting HeLa cells with LNA Gapmers and harvesting them after 24 or 48 h for RNA-seq or ATAC-seq (Figure 1A). The time-resolved MALAT1 knockdown (KD) effect was validated by real-time PCR (Figure 1B) and RNA-seq (Figure 1C) at both timepoints, where a Gapmer against Beta Galactosidase (LacZ) was used as a negative control. The genome-wide transcriptional changes showed that 24 h after MALAT1 KD, there was a greater number of downregulated transcripts (275) than upregulated transcripts (124) (Figure 1D,E; Appendix A). The vast majority of the differentially expressed transcripts were protein-coding genes (Figure 1E, Appendix A). At 48 h after MALAT1 Gapmer transfection, the number of differentially expressed transcripts strongly increased, reaching similar levels for both up- and downregulated transcripts (Figure 1D,E; Appendix A). At this timepoint, in addition to the prevalence of protein-coding genes, we also observed the differential expression of non-coding genes (Figure 1E and Appendix A). Gene ontology analysis by overrepresentation test [27,28] of the differentially expressed transcripts at 24 h after MALAT1 KD revealed that the downregulated genes were enriched in transcripts associated with transcriptional regulation (ATP-dependent activity, acting on DNA, and transcription coregulator activity) (Figure 1F, left). While upregulated genes were enriched in transcripts associated with the structural components of ribosomes and chromatin (Figure 1F, right).

The differences observed between 24 and 48 h after MALAT1 KD suggest two important aspects of the function of this lncRNA. First, MALAT1 is predominantly associated with the regulation of protein-coding genes. Second, the time-dependent changes in the transcriptome suggested the presence of primary target genes after 24 h and the emergence of incidental targets 48 h after knockdown, when transcriptional regulators (proteins and nc-RNAs) become differentially expressed.

### 2.2. Chromatin Accessibility Changes after MALAT1 Downregulation

To delineate potential mechanisms by which MALAT1 regulates the expression of protein-coding genes through changes in chromatin accessibility, we performed ATAC-seq experiments 24 and 48 h after MALAT1 KD (Figure 1A). In general, after MALAT1 KD, sites became accessible (Figure 2A, Clusters 1–3) or inaccessible (Figure 2A, Cluster 4) after 24 h, and the magnitude of the change tended to increase at 48 h. Therefore, considering that MALAT1 primary target genes were found after 24 h of KD via RNA-seq, we focused on early structural changes at this time point. We identified numerous significantly altered ATAC sites with increased or decreased chromatin accessibility distributed across all chromosomes (Appendix A). Nevertheless, by comparing against genomic sites, we calculated the enrichment or reduction of the significantly altered ATAC sites per chromosome. The analysis revealed enrichment of significantly altered ATAC sites on chromosomes 1, 5, 6, 8, 11, 12, 15, 17, and 20 and reductions in chromosomes 4, 9, 13, 14, 16, 21, and X (Appendix A). Despite the broad changes in accessibility, these changes were concentrated on specific chromosomes. 

Then, to determine whether the significant changes in accessibility were located within REs, we analyzed their enrichment over specific genomic elements and found that enhancer regions are the most affected after 24 and 48 h after MALAT1 KD (Figure 2B). Thus, to understand the accessibility changes in terms of gene expression, we defined two scenarios for MALAT1 gene regulation through accessibility changes. The first one was with MALAT1 inducing the expression of a target gene maintaining the accessibility of the corresponding REs under normal conditions (scenario 1), and the second one was where MALAT1 represses the expression of a target gene by decreasing the accessibility of the corresponding REs under normal conditions (scenario 2) (Appendix A). Using both scenarios as parameters to find the overlap between differentially expressed genes (DEGs) and differentially accessible regions (DARs), we found a small number of genes up- or downregulated according to the accessibility changes after MALAT1 KD (Figure 2C). These results indicate that transcriptional changes following accessibility alterations due to the loss of MALAT1 are limited to specific genes. KEGG co-expression analysis showed that pathways related to cancer are the most enriched among DEGs and DARs together (Figure 2D), where nuclear glucocorticoid receptor binding is the only significant enriched GO term found for scenario 1 (with ETS2 and NR4A1 genes), while for scenario 2, we found more terms that are significantly enriched; nevertheless, they are more general compared to scenario 1 (Appendix A).

Now, examining the sites that gained or lost accessibility, we focused on the two sites exhibiting the greatest extent of decrease in accessibility after 24 h of MALAT1 KD (Figure 2E, dots inside the blue oval). These sites are part of cluster 4 (Figure 2A) and are located on chromosome 12, a chromosome exhibiting a higher percentage of inaccessible sites than average (Figure 2F).

### 2.3. MALAT1 Regulates NR4A1 Expression by Modulating the Chromatin Accessibility of a Downstream RE

Specifically, the highly inaccessible sites found after MALAT1 KD were located downstream of the NR4A1 gene (Chr12: 52.022.832-52.059.507 on GRCh38) in a region flanked by additional strong ATAC peaks and coinciding with CTCF-ChIP-seq peaks (red and black squares, respectively; Appendix A). This finding suggested the presence of a topologically associated domain (TAD), in which the NR4A1 and ATG101 genes are expressed, as observed by our RNA-seq data (Appendix A). Additionally, the active histone marks H3K27Ac/H3K4me3/H3K4me1 indicate the NR4A1 and ATG101 promoters and enhancer regions upstream and downstream of the genes, respectively (Figure 3A). Cap analysis of gene expression (CAGE) data from HeLa cells revealed the location of the transcriptional start site, corresponding to the NR4A1 transcript ENST00000394825 (Appendix A). The promoter and coding region of this transcript cover a broad ATAC-accessible region, including the downstream element (Figure 3A).

Twenty-four hours after MALAT1 KD, the *NR4A1* gene was significantly downregulated without affecting the expression of ATG101 (Figure 1D, Appendix A). This effect was verified by RT–qPCR (Figure 3B), which suggested a primary MALAT1-dependent loss of chromatin accessibility at a novel NR4A1-downstream RE (Figure 3A, black box). To test the functionality of the proposed RE, we used the CRISPR-i system, which consists of catalytically inactive Cas9 (dCas9) and three independent guide RNAs (gRNAs) targeting the NR4A1-downstream RE (Figure 3C). Guide RNAs were designed to bind directly to the proposed RE (gRNA2) or 1 kbp (gRNA1) and 2 kbp (gRNA3) downstream of the RE. HeLa cells were independently transfected with plasmids encoding guide RNAs (Appendix A) and dCas9. RT–qPCR analysis revealed the repression of NR4A1 expression by all three guide RNAs, with gRNA 2 directly targeting the NR4A1-downstream RE, being the most efficient at repressing NR4A1 expression (Figure 3C). These results suggest the presence of a potential downstream regulatory element for the *NR4A1* gene.

Additionally, we tested whether the MALAT1 lncRNA is physically associated with the NR4A1-downstream RE to maintain accessibility at this locus. Using the RAP assay, we pulled-down MALAT1 from HeLa nuclear extracts (Appendix A). However, we did not find a direct physical interaction between MALAT1 and the NR4A1-downstream RE (Appendix A), suggesting that additional factors mediate the chromatin accessibility of this locus (see discussion).

Taken together, the results showed that MALAT1 regulates the accessibility of a novel downstream regulatory element downstream of the NR4A1 gene, fine-tuning its expression in HeLa cells.

### 2.4. The Accessibility of the NR4A1-Downstream RE Correlates with NR4A1 Expression in Specific Cancer Types

To address the functional role of the NR4A1-downstream RE, we screened the TCGA cancer database [29]. Indeed, we observed an overlap between the NR4A1-downstream RE and two ATAC-seq peaks derived from TCGA data (Figure 4A, PEAK1 and PEAK2). The intensity of both ATAC-seq peaks was positively correlated with NR4A1 expression in all cancer types (Figure 4B). Furthermore, analyzing the accessibility of the two ATAC peaks in individual cancer types showed that the accessibility of this region was cancer type-specific (Appendix A). The highest level of accessibility was observed for pheochromocytoma and paraganglioma (PCPG), and the lowest ATAC-seq scores were found for kidney renal papillary cell carcinoma (KIRP) (Appendix A). These cancer-specific effects led us to evaluate the correlation between accessibility for both peaks and NR4A1 expression in different cancer types. Here, we identified a strong correlation between breast invasive carcinoma (BRCA) and other cancer types (Figure 4C).

The direct correlation between NR4A1 expression and the accessibility of the NR4A1-downstream RE defined a MALAT1/NR4A1 axis, which, according to our initial results in HeLa cells and the bioinformatic analysis of TCGA data, suggested that MALAT1 regulates NR4A1 expression through accessibility changes in the NR4A1-downstream RE in a cancer-specific manner.

### 2.5. The MALAT1/NR4A1 Axis Is Functional in Breast Cancer Cells but Not in Pancreatic Cancer Cells

To test the hypothesis that a cancer-type-specific MALAT1/NR4A1 axis modulates the expression of NR4A1, we evaluated the effect of this axis in specific cancer cell types. We tested whether MALAT1 expression and NR4A1-downstream RE accessibility were strongly correlated in cancer types, such as BRCA (Figure 4C), versus low- or no-correlation cancers, such as pancreatic adenocarcinoma (PDAC) (Figure 5A). To assess the functionality of the MALAT1/NR4A1 axis, we transfected MCF7 breast cancer and PANC1 pancreatic adenocarcinoma cells with Gapmers to downregulate MALAT1 expression (Figure 5B). MALAT1 KD induced a significant decrease in NR4A1 expression in MCF7 cells, while no difference was observed in PANC1 cells compared to the controls (Figure 5C). Furthermore, we evaluated the regulatory function of the NR4A1-downstream RE by transfecting dCas9 and gRNA2 into MCF7 and PANC1 cells (Figure 5D) and evaluated the expression of NR4A1. A significant reduction in the expression of NR4A1 was observed in MCF7 cells but not in PANC1 cells (Figure 5E). These results indicate that the NR4A1-downstream RE is sufficient to modulate the expression of the NR4A1 gene in MCF7 cells but is not functional in PANC1 cells.

## 3. Discussion

LncRNAs are key players in the regulation of cellular transcription. During cancer onset and progression, altered transcriptional programs trigger uncontrolled intracellular signaling that normally induces cell dedifferentiation and proliferation [30,31,32]. In this context, the lncRNA MALAT1 is involved in different cancer types and acts as a transcriptional regulator of oncogenes and tumor suppressors [9,19,20]. Our work focused on partially deciphering the MALAT1-dependent molecular mechanism involved in gene expression regulation by altering the chromatin accessibility of primary gene targets.

### 3.1. MALAT1-Dependent Changes in Chromatin Accessibility

The time-resolved downregulation of MALAT1 demonstrated that its primary target genes corresponded mainly to protein-coding genes. MALAT1 KD induced alterations in the chromatin accessibility of a large number of loci. Such an increase or decrease in chromatin accessibility after the KD of a lncRNA has been described previously [33,34,35,36], confirming that lncRNAs are active chromatin regulators. Nevertheless, to our knowledge, no study has investigated the impact of MALAT1 on chromatin accessibility. Our ATAC-seq data revealed the largest changes in chromatin accessibility downstream of the *NR4A1* gene after MALAT1 KD. The local decrease within an open 36 kbp region was associated with the repression of *NR4A1* gene expression. We further showed that the accessibility of this downstream element was MALAT1-dependent, where CRISPR-i experiments targeting the NR4A1-downstream RE demonstrated the presence of a regulatory element for the *NR4A1* gene. Similar types of REs have been reported to regulate the Cja1 (Cx43) and mouse globin genes [37,38]. This led us to think about a possible interaction between MALAT1 and the NR4A1-downstream RE locus. Nevertheless, MALAT1 RAP assays could not detect the suggested physical interaction, indicating that MALAT1 is not the sole factor involved in the accessibility maintenance of the NR4A1-downstream RE in HeLa cells. Consistent with these findings, MALAT1 CHART-seq data showing direct MALAT1-chromatin contacts did not reveal any interaction with the NR4A1-downstream RE in MCF7 breast cancer cells, confirming our MALAT1 RAP findings in HeLa cells (Appendix A) [18]. Our ATAC-seq data showed that the NR4A1 promoter and the downstream RE correspond to the region with the highest accessibility of the whole chromosome 12. Therefore, the decrease in the accessibility of the NR4A1-downstream RE after MALAT1 KD must involve additional factors that maintain the accessibility of this region under normal conditions. Data retrieved from ChIP-atlas (https://chip-atlas.org/ accessed on 28 December 2021) in HeLa cells revealed that the transcription activator BRG1, the catalytic subunit of the SWI/SNIF chromatin remodeling complex, binds to the whole *NR4A1* gene body, including the NR4A1-downstream RE (Appendix A). A direct MALAT1-BRG1 interaction was previously shown to increase the inflammatory response in hepatocellular carcinoma [39]. Moreover, two other transcription factors involved in the super-elongation complex of RNA polymerase II, AFF4 and ELL2 [40], also bind to the *NR4A1* gene body and the NR4A1-downstream RE (Appendix A). MALAT1 KD resulted in a minor decrease in BRG1 and AFF4 expression but also in the significant downregulation of ELL2 (Appendix A). Therefore, the putative MALAT1-BRG1 interaction, the significant decrease in ELL2 expression after MALAT1 KD, and the AFF4 occupancy over the RE could explain the MALAT1-dependent changes in chromatin accessibility at the NR4A1-downstream RE region (Figure 5F). Nevertheless, in addition to the additional protein factors, we cannot rule out the possibility that other ncRNAs are involved in maintaining the accessibility of this region.

Interestingly, the expression of an antisense lncRNA against NR4A1, NR4A1AS, was shown to modulate NR4A1 expression in colorectal cancer cells [41]. The promoter region of NR4A1AS1 coincides with the NR4A1-downstream RE found in this study, suggesting that the expression of this lncRNA is an interesting candidate for understanding the changes in NR4A1-downstream RE accessibility. However, our RNA-seq data did not reveal the presence of this lncRNA in HeLa cells. Moreover, these results were verified by RT–qPCR assays in HeLa and MCF7 cells, in which no expression of the lncRNA NR4A1AS was detected. These results indicate that NR4A1AS is not involved in the model systems used in this study but likely plays a tissue-specific role in other cellular models.

### 3.2. The Impact of the MALAT1/NR4A1 Axis on Breast Cancer Development

Analysis of TCGA data revealed that the chromatin accessibility of the NR4A1-downstream RE is positively correlated with NR4A1 expression in specific cancer types, as we observed in HeLa cells. Here, we focused on breast carcinoma cells due to their high correlation. We recapitulated the MALAT1/NR4A1 axis in ER-positive MCF7 breast cancer cells but did not find this axis in pancreatic adenocarcinoma PANC1 cells, as predicted by the lack of correlation in PDAC analyses. Additionally, this cell type-specific effect could be explained by mutations on the NR4A1-downstream RE or the TAD boundaries where this gene is located on the pancreatic cancer cells, but not on the breast cancer cells, following the enhancer retargeting process observed on the ZCCHC7 gene during B cell lymphoma progression [42].

The function of MALAT1 in breast cancer cells is still a matter of debate. Both, oncogenic and tumor suppressor roles have been described for this lncRNA [43]. Our results agree with the role of MALAT1 as a tumor suppressor maintaining NR4A1 expression in MCF7 cells. On one hand, NR4A1 was shown to upregulate ERK signaling at low levels and preserve the levels of proapoptotic proteins, regulating proliferation and controlled cell death [26]. Through these mechanisms, NR4A1 seems to maintain the tamoxifen sensitivity of MCF7 cells [26]. In contrast, in triple negative breast cancer (TNBC) MDA-MB-231 cells, NR4A1 promotes tumor invasion and metastasis by activating transforming growth factor beta (TGF-R) signaling [25]. These results suggest that, depending on the cellular environment, the MALAT1-NR4A1 axis can function as a tumor promoter or suppressor (Figure 5F).

The transcription factor NR4A1 has been described as an immediate early gene (IEG) that corresponds to a group of genes that rapidly respond to stress through their fast and transient transcriptional stimulation [44]. Replication stress in the absence of checkpoints and quality control machinery is one of the first steps leading to genomic instability during cancer development [45]. By comparing non-malignant MCF10 breast epithelial cells with TNBC MDA-MB-231 cells and circulating patient-derived breast tumor cells, NR4A1 was recently reported to modulate IEG expression through interaction with gene bodies and suppression of transcriptional elongation. Under oncogenic replication stress conditions, NR4A1 is released, triggering a burst of IEG expression [46].

Taken together, these data point to a mechanism by which MALAT1 modulates the accessibility of an NR4A1-downstream RE, probably through other protein factors, regulating NR4A1 expression levels. The presence of MALAT1 seems to fine-tune NR4A1 expression levels, which in turn are associated with low proliferative signaling and pro-apoptosis cues in ER-positive MCF7 cells. The discovery of the MALAT1-NR4A1 axis increases the potential of these genes as therapeutic targets for ER-positive breast cancer and drug resistance treatments.

## 4. Materials and Methods

### 4.1. Cell Culture

HeLa, MCF7, and PANC1 cells were cultured in HyClone Dubelco’s modified Eagle medium (DMEM) supplemented with glucose (4.5 g/L), L-glutamine (584 mg/L), sodium pyruvate (110 mg/L), and 10% fetal bovine serum. The cells were maintained at 37 °C in 5% CO_2_ and subcultured using 0.25% trypsin.

### 4.2. Western Blot

Protein extracts from HeLa and MCF7 cells were obtained after homogenization in RIPA buffer supplemented with protease inhibitors by pipetting and sonication at 4 °C. The proteins were resolved by SDS-PAGE and transferred to PVDF membranes. The membranes were blocked with 5% milk in TBST (TBS with 0.1% Tween-20^®^), probed with antibodies against GAPDH and the HA-tag, and then with horseradish peroxidase-coupled secondary antibodies. Luminescent blot signals were obtained by incubating the samples with peroxidase substrate and imaged on a Syngene G:box gel documentation system.

### 4.3. Real-Time PCR

Total RNA (1 µg) from HeLa, MCF7, and PANC1 cells was used for cDNA preparation with a RevertAid RT Kit (Thermo Scientific, Waltham, MA, USA). Real-time PCRs were prepared in 0.1 mL tubes with a final volume of 20 µL containing specific primers (MALAT1, NR4A1, and GAPDH), variable amounts of cDNA (depending on the target), 10 µL of the 2X JumpStart^TM^ ReadyMix^TM^, 1 mM MgCl_2_, and SyberGreen (1:400,000 from the 10,000X stock, Thermo Scientific, Waltham, MA, USA). Reactions were analyzed on a Rotor-Gene RG3000 thermal cycler.

### 4.4. Cloning of Guide RNAs

The backbone vector (pZDonor) was modified to include BbsI restriction sites, generating the pSPgRNA plasmid (#47108 Addgene), to allow the cloning of gRNAs of choice as described before [47]. In brief, guide RNAs were ordered as oligonucleotides from IDT Technologies (Figure 3E), hybridized, phosphorylated, ligated with the BbSI digested pSPgRNA plasmid, and cloned into DH5α cells. Positive clones were verified initially by colony PCR and then by sequencing (Appendix A).

### 4.5. Transfections

HeLa cells were reverse-transfected with Lipofectamine 2000 for Gapmer-mediated MALAT1 downregulation and CRISPR-I assays. Gapmer transfections were carried out using 1.5 µL of 200 µM Gapmers against MALAT1 or LacZ and 5 µL of Lipofectamine for 1.2 million cells in a 6-well plate for 24 h post transfection. For 48 h post transfection, we used half the number of cells. For CRISPR-i, we used 5 µL of Lipofectamine, 750 µg of dCas9 plasmid (#61355 Addgene), and 750 µg of each gRNA plasmid (#47108 Addgene) for 0.5 million cells on a 6-well plate. The cells were harvested 48 h post transfection.

MCF7 cells were plated and transfected with Lipofectamine 3000 for Gapmer-mediated MALAT1 downregulation and CRISPR-I assays. Gapmer transfections were carried out using 2 µL of 200 µM Gapmers, against MALAT1 or LacZ, 2 µL of lipofectamine, and 1 µL of P3000 reagent; 0.5 million cells were seeded on a 6-well plate the day before transfection. Cells were harvested 24 h post transfection. For CRISPR-i, we used 2 µL of Lipofectamine, 650 µg of dCas9 plasmid (#47106 Addgene), 650 µg of each gRNA plasmid (#47108 Addgene), and 2.5 µL of P3000 reagent for 0.5 million cells seeded the day before they were seeded on a 6-well plate. The cells were harvested 48 h post transfection.

### 4.6. Custom Genome

A custom version of the hg38 human genome was used for all sequencing analyses, in which all repeats of ribosomal genes were masked with N bases and a single repeat of the ribosomal gene was kept as an extra chromosome named “ChrR” [48].

### 4.7. Annotation

A custom version of the gencode v27 *Homo sapiens* annotation was used, with ribosomal RNA present one time on an extra chromosome, ChrR, to fit our custom genome.

### 4.8. RNA Purification

Total RNA purification for cDNA preparation was performed following the manufacturer’s instructions for TRIzol^®^ Reagent (Ambion by Life Technologies, Carlsbad, CA, USA). For ChRIP assays, RNA purified from immunoprecipitated samples was extracted by resuspending the beads in proteinase K buffer (10% SDS, 50 mM EDTA, 10 mM Tris-Cl, pH 7.4) and incubating them with 50 µg of proteinase K for 1 h at 50 °C with gentle agitation. Then, the crosslinking was reversed at 65 °C for 2 h, followed by RNA purification with TRIzol and isopropanol precipitation.

### 4.9. RNA Sequencing and Transcriptome Analysis

The quality and adapter contamination of the raw files were first checked with fastqc v0.11.9 and multiqc v1.13 [49,50]. Following the quality check, adapters were removed using Trimmomatic v0.38 [51], with the following flags: PE -phred33 ILLUMINACLIP:TruSeq3-PE-2.fa:2:30:10 LEADING:3 TRAILING:3 MINLEN:30 SLIDINGWINDOW:4:20 and cutadapt v4.0 [52] for library- and sequencer-specific trimming, with the following flags: -u 5-a “A{100}” -a “G{100}” -a “T{100}”. The reads were then aligned to the reference genome hg38 using STAR v2.7.2a [53] and the following flags: --sjdbGTFfile gencode.v27.annotation.gtf --outFilterType BySJout --outFilterMultimapNmax 20 --alignSJoverhangMin 8 --alignSJDBoverhangMin 1 --outFilterMismatchNoverReadLmax 0.04 --alignIntronMin 20 --alignIntronMax 1000000 --alignMatesGapMax 1000000 --outSAMtype BAM SortedByCoordinate --outFilterScoreMinOverLread 0 --outFilterMatchNminOverLread 0 --outFilterMatchNmin 0. Unmapped reads and reads with a MAPQ mapping quality lower than 30 were removed for downstream analysis using SAMtools v1.90 [54]. Next, read counts were obtained with FeatureCount from the Subread package v2.0.0 [55] using the following flags: -p -s 1 -t exon -g gene_id -C -B. Differential expression analysis between the knockdown samples and the controls was performed in R v4.1.2 with the DESeq2 R package [56], with apeglm shrinkage estimation for ranking of genes and visualization [57]. Differentially expressed genes were filtered with an absolute log fold change > 1.0 and an FDR < 0.05.

### 4.10. ATAC-seq Assays, Sequencing, and Analysis

The ATAC assay was performed following the instructions of the Omni-ATAC protocol [58] using the Nextera DNA Flex Library prep kit (Illumina, San Diego, CA, USA). Tagmented DNA was analyzed on a bioanalyzer and sequenced on a Next-Seq 550 device in the Genomics core unit of the Regensburger Centrum für Interventionelle Immunologie.

The quality and adapter contamination of the raw files were first checked with fastqc v0.11.9 and multiqc v1.13 [49,50]. Following the quality check, adapters were removed using Trimmomatic v0.38 [51]. Paired-end reads were then mapped to the reference genome hg38 using Bowtie2 [59] with the following flags: --local --very-sensitive-local -X 5000 --no-discordant -dovetail. Unmapped reads and reads with a MAPQ mapping quality lower than 30 were removed for downstream analysis using SAMtools v1.90 [54]. Reads mapping to chrM, blacklisted regions, and PCR duplicates were removed using SAMtools and Picard, respectively. Finally, the reads were shifted based on the Tn5 cuts using DeepTools v3.1.2 [60] alignmentSieve--ATACshift. Peak calling was performed using MACS2 v2.2.71 [61], with the following flags: callpeak -f BAMPE -g hs -broad -keep-dup all -buffer-size = 100. Differentially accessible regions were obtained with the DiffBind R package. Differentially accessible regions were filtered with an absolute Log fold change > 1.0 and an FDR < 0.01. Regulatory regions were assigned to genes using the GREAT tool [62] via the basal plus extension method (5 kb upstream, 1 kb downstream plus distal up to 50 kb).

### 4.11. Correlations with TCGA and Pancreatic Cancer Data

The normalized ATAC-seq peak signals (ID: Peak1 = PRAD_72732 and Peak2 = PRAD_72733) of TCGA tumor samples were obtained from the ATAC-seq Hub (https://atacseq.xenahubs.net accessed on 6 February 2021) [29]. The corresponding gene expression data of the matching donors were quantified using the TCGAbiolinks Bioconductor package [63]. Here, we used the upper quartile-normalized FPKM values.

The accessibility signals for Peak1 and Peak2 were subsequently plotted against the NR4A1 expression. For the PANC1 cell line, ATAC-seq and RNA-seq data were obtained from the Gene Expression Omnibus (GEO) database (GSE124229 and GSE124231). First, the gene expression and accessibility signals were normalized to the library size for each patient. The accessibility signals for Peak1 and Peak2 were subsequently plotted against the NR4A1 expression. Linear regression was used to model the correlation between gene accessibility and gene expression signals in R v.4.1.2. Heatmaps were generated using DeepTools v.3.4.3 [60] and the pheatmap R package.

## Figures and Tables

**Figure 1 ijms-25-05515-f001:**
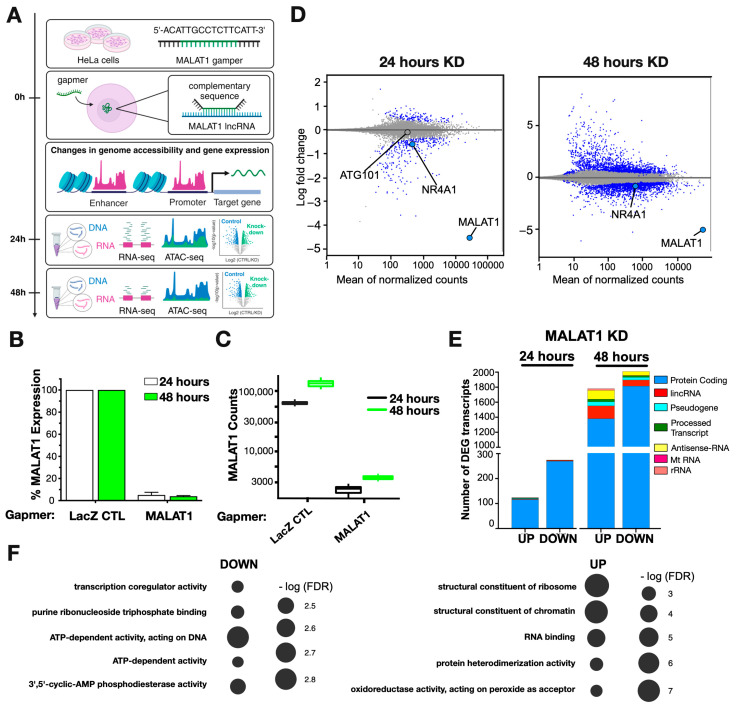
MALAT1 primarily modulates the expression of protein-coding genes. (**A**) Methodology for determining the time-resolved MALAT1 KD through the transfection of Gapmers, followed by RNA-seq and ATAC-seq assays in HeLa cells. Validation of MALAT1 KD at 24 and 48 h post transfection by real-time PCR (**B**) and RNA sequencing (**C**). (**D**) MA plot of the RNA-seq data showing the MALAT1 KD values at 24 and 48 h post transfection. Significantly up- and downregulated transcripts are indicated in blue. (**E**) Quantification of significantly up- and downregulated MALAT1 transcripts from RNA-seq data at 24 and 48 h post transfection. (**F**) Gene ontology analysis by overrepresentation test of the differentially expressed transcripts 24 h post transfection.

**Figure 2 ijms-25-05515-f002:**
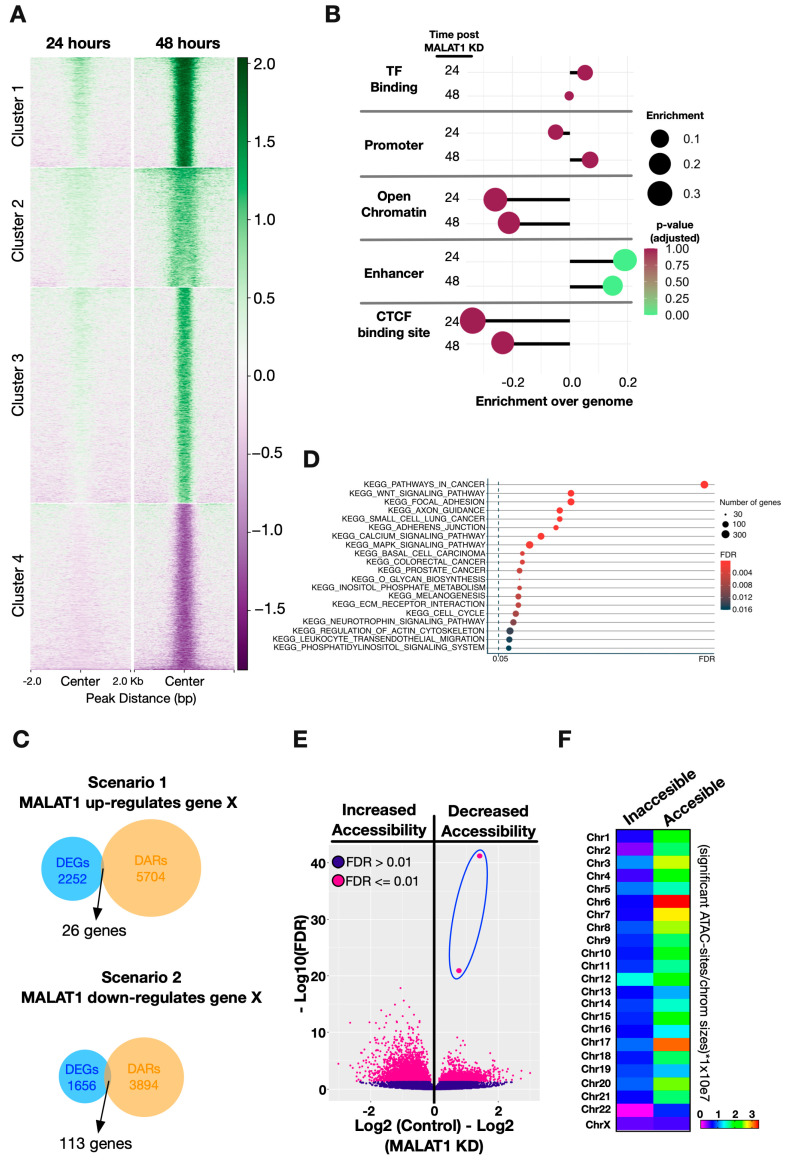
Chromatin accessibility changes after MALAT1 downregulation. (**A**) Heatmap showing the K-means clusters of differentially accessible ATAC-seq sites in MALAT1 KD versus control 24 and 48 h post transfection. Accessibility scores are shown on the right side. (**B**) Enrichment of significant ATAC sites over specific regulatory elements after 24 and 48 h post MALAT1 KD. (**C**) Overlap quantification of the differentially expressed genes (DEGs) and differentially accessible regions (DARs) after 24 and 48 h post MALAT1 KD, according to the scenarios described on the Appendix A. (**D**) KEEG co-expression analysis of the DEGs and genes associated with DARs after MALAT1 KD. (**E**) Extent and type of change in chromatin accessibility after MALAT1 KD for significant (purple dots) and non-significant (violet dots) ATAC-seq sites. (**F**) Chromosomal distribution of significant accessible/inaccessible ATAC-seq sites after MALAT1 KD.

**Figure 3 ijms-25-05515-f003:**
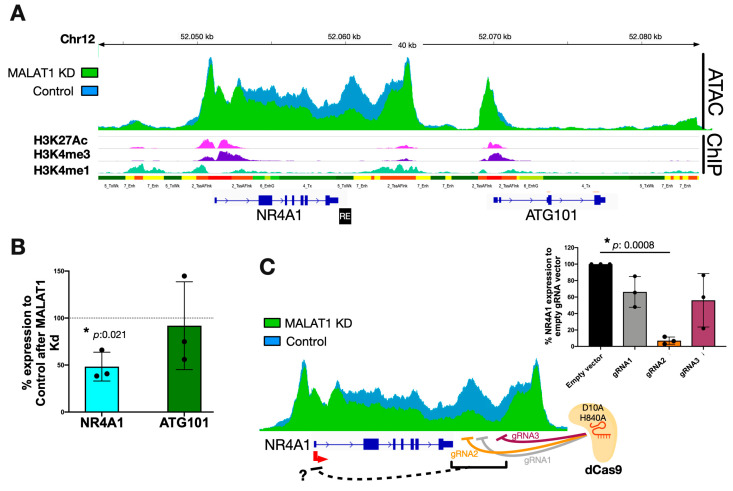
MALAT1 modulates the accessibility of a regulatory element downstream of *NR4A1*. (**A**) Chromatin states throughout the NR4A1 gene. (**B**) Real-time PCR expression analysis of NR4A1 and ATG101 after MALAT1 KD. (**C**) Scheme depicting the CRISPR-i approach using 3 gRNAs against the *NR4A1*-downstream regulatory element. Real-time PCR quantification of NR4A1 expression after CRISPR-i assays. * Significant differences are shown with *p*-values obtained from *t* tests.

**Figure 4 ijms-25-05515-f004:**
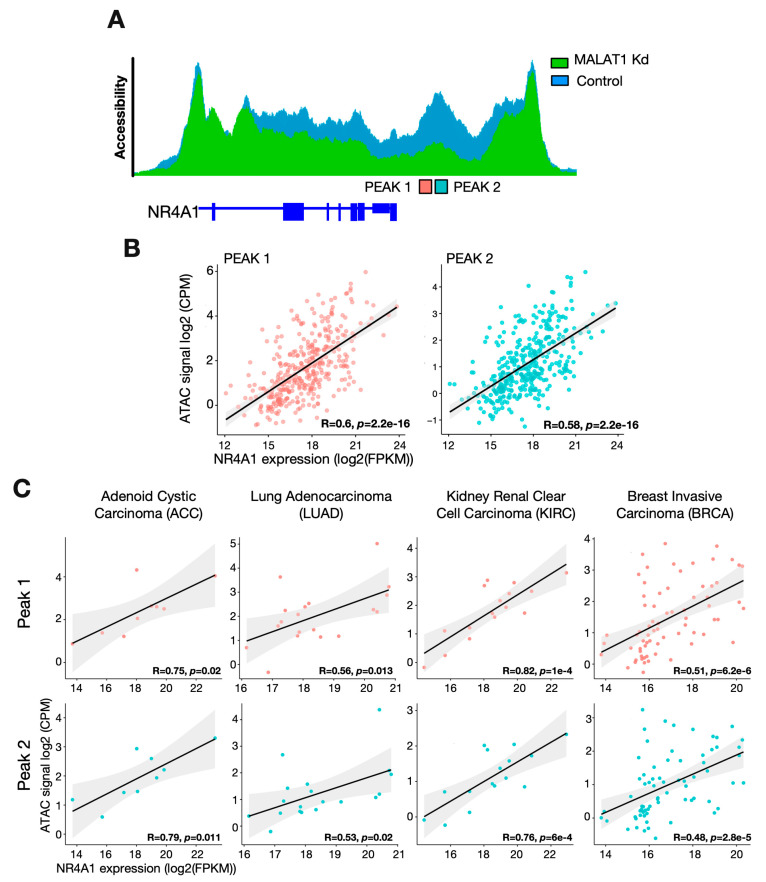
Cancer-specific correlation between the accessibility of the NR4A1-downstream RE and the expression of NR4A1. (**A**) Schematic representation of the TCGA-derived peaks (PEAK1 and PEAK2) found on the NR4A1-downstream RE. (**B**) Correlation of TCGA-derived peak accessibility and NR4A1 expression in all TCGA cancer types shown in Appendix A. (**C**) Correlations of the accessibility of TCGA-derived peaks with NR4A1-downstream RE and NR4A1 expression in individual TCGA cancer types.

**Figure 5 ijms-25-05515-f005:**
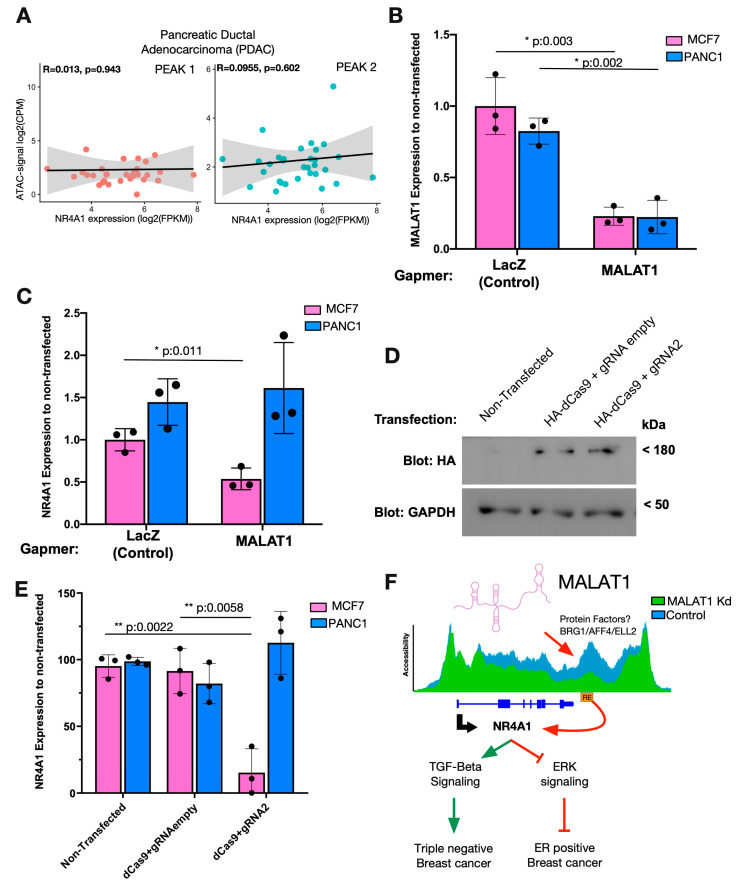
MALAT1 regulates NR4A1 in breast cancer cells but not in pancreatic adenocarcinoma cells. (**A**) Correlation between TCGA-derived peak accessibility and NR4A1 expression in PDAC patients. (**B**) MALAT1 KD validation in MCF7 and PANC1 cells using Gapmers. (**C**) Real-time PCR analysis of NR4A1 expression after MALAT1 KD in MCF7 and PANC1 cells. (**D**) Western blot analysis of HA-tag expression to verify the effect of dCas9 transfection on MCF7 cells. (**E**) CRISPR-i assays with gRNA2 analyzing the expression of NR4A1 by real-time PCR in MCF7 and PANC1 cells. (**F**) Scheme representing the molecular mechanism underlying the MALAT1-mediated modulation of NR4A1 through the accessibility of a downstream RE. * Significant differences are shown with *p*-values obtained from *t* tests.

## Data Availability

ChIP data for the histone marks H3K27Ac, H3K4me3, and CTCF were extracted from ENCODE under the accession GSE29611. The raw sequencing data were deposited in The Sequence Read Archive (SRA) under the accession ID PRJNA943483. The code used for the analysis is available on GitHub: https://github.com/sarawernig/Cancer-specific-MALAT1-regulation.

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
