# Peer review of "The Long Non-Coding RNA MALAT1 Modulates NR4A1 Expression through a Downstream Regulatory Element in Specific Cancer Cell Types"

_ijms, 2024, doi:10.3390/ijms25105515_

Round 1
Reviewer 1 Report
Comments and Suggestions for Authors
The manuscript entitled “The long non-coding RNA MALAT1 modulates NR4A1 expression through a downstream regulatory element in specific cancer cell types” submitted by Zorc et.al. discusses the regulation of NR4A1 by the MALAT1. The authors showed a mechanistic model by which the MALAT1 modulated the regulatory elements necessary for the expression of NR4A1 by protein factors in the breast cancer and not in other cancer models. The authors further showed that this regulation results in the defects in the proliferative signals and the pro-apoptotic signals which was shown in the breast cancer cell line MCF7. Finally, the authors proposed that this regulation can be used for potential targeting for ER-positive breast cancer and drug resistant.
The authors have a clear hypothesis and to test this the author used appropriate experimental design. The manuscript is well written, and the statistical methods used are appropriate to the best of my knowledge. The discussion is well written. However, there are some concerns as follows.
For the Results 2.1 there is mentioning of Figure 1F but in the figures there is no F panel for Figure 1
For the figure 2 A heat map the authors should mention if the center is the gene center or is this the TSS
For the results section 2.1 the authors should mention few known targets which are known to be changed and obtained in their DGE and cite them.
Similarly, for the ATAC seq data mentioned in 2.2 have the authors seen any know binding sites for MALAT1 like CALR and RAD23A shown in (PMID: 25155612) and cite them . If so, they should show mention them and show qPCR data for the same.
The authors should show if the gRNA2 has off target sites.
For the results section 2.5 the authors should show the use of dCas9 against the RE not only affects the RNA levels but also affects the protein levels of NR4A1 by western blotting.
In the discussion section the authors mentioned the RE changes the accessibility for the NR4A1. It is worthwhile to mention and cite other possible mechanism like enhancer retargeting in cancer cells by mutations in these RE as shown in PMID: 38049665.
Overall, the work done by Zorc et.al is commendable.
Reviewer 2 Report
Comments and Suggestions for Authors
The authors investigate how MALAT1 lncRNA regulates target genes via time-resolved knockdowns and RNA-seq/ATAC-seq assays. Their results indicate that MALAT1 primarily controls specific protein-coding genes, notably reducing chromatin accessibility downstream of the NR4A1 gene, thereby decreasing its expression. TCGA data showed a positive correlation between NR4A1 expression and its downstream accessibility in breast cancer, but not in pancreatic cancer. Therefore, the results suggest that MALAT1 is involved in fine-tuning NR4A1 expression in a cell type-specific manner.
This study provides important insights into the transcriptional regulatory roles and mechanisms of the lncRNA MALAT1, offering valuable contributions to the field. The experimental procedures and results are well-executed, demonstrating the impact of MALAT1 as a transcriptional regulator across different cancer types. However, some revisions are needed to enhance the clarity and presentation of the findings:
1.Structure of the manuscript:
The paper currently lacks a 'Conclusions' section, which is critical for summarizing the findings and implications of the research. Additionally, the 'Materials and Methods' section should be placed immediately after the 'Introduction' to maintain the logical flow of the manuscript.
2.Revision of Figures 2-4:
There are several issues with the current figures that must be addressed:
・The data in these figures are not clearly verifiable as some axis titles are not visible, and the resolution of the figures is not sufficient.
・The fonts used within the figures are inconsistent, which can distract from the data presentation.
・Consider moving less critical data to the Supporting Information to streamline the main text and focus on the most impactful results.
These structural and graphical enhancements are crucial for ensuring that the manuscript effectively communicates the authors' significant findings. By addressing these points, the paper will likely become a more impactful and clearer resource for researchers in this field.
